# Emotion dynamics in movie dialogues

**Will E. Hipson** [1]*, **Saif M. Mohammad** [2]*

**1** Carleton University, Ottawa, ON, Canada, **2** National Research Council Canada, Ottawa, ON, Canada

* william.hipson@carleton.ca (WEH); saif.mohammad@nrc-cnrc.gc.ca (SMM)

## Abstract

Emotion dynamics is a framework for measuring how an *individual's* emotions change over time. It is a powerful tool for understanding how we behave and interact with the world. In this paper, we introduce a framework to track emotion dynamics through one's utterances. Specifically we introduce a number of *utterance emotion dynamics (UED)* metrics inspired by work in Psychology. We use this approach to trace emotional arcs of movie characters. We analyze thousands of such character arcs to test hypotheses that inform our broader understanding of stories. Notably, we show that there is a tendency for characters to use increasingly more negative words and become increasingly emotionally discordant with each other until about 90% of the narrative length. UED also has applications in behavior studies, social sciences, and public health.

**Data Availability Statement:** Public repo: https://github.com/whipson/edyn.

**Funding:** The author(s) received no specific funding for this work.

## Introduction

We often think of an emotion as a fleeting response to some event: we feel joy when we see a childhood friend, or anger when slighted by a colleague. However, our emotions are more complex. We are always in some emotional state (even if this state is relatively neutral), and there is mounting interest in understanding how our emotional state changes over time and how these changes differ among different people [1].

In Psychology, emotion dynamics is the study of patterns of change and regularity in emotion [2, 3]. Researchers use *intensive longitudinal data* (repeated self-reports separated by relatively short time intervals [4]) to describe and/or predict how a person's emotions change over time.

Self-reports offer a unique window into a person's emotional state, but they are only a proxy of actual feelings. An alternative window into our emotions is through the words we use. If we are happy, we are likely to utter more happiness-associated words than usual, if we are angry, we are likely to utter more anger-associated words, and so on [5].

One source of abundant intensive longitudinal text is character dialogue from literature and film. Character dialogue drives the movie's plot and is the most direct way through which the audience can understand what a character is thinking and feeling. Past attempts to model emotions in story dialogue have paid little attention to *individual* characters and their emotion words, instead averaging across narrative text and character dialogue [6, 7].

In this paper we introduce (Psychology inspired) metrics of emotions dynamics derived from one's utterances—*Utterance Emotion Dynamics (UED)*, such as *home base* (typical

**Competing interests:** The authors have declared that no competing interests exist.

emotional state), *variability* (unpredictability), and *rise/recovery rates* (emotional reactivity/regulation). We explored these UED metrics on a corpus of movie dialogues to better understand characters' use of emotion words and, in turn, obtain a deeper perspective of emotions in fictional narratives.

The basic component driving UED metrics are the emotion associations of the individual words in the utterances. Although a sentence has more meaning than the aggregate of the meanings of its constituent words, automatic emotion detection from whole sentences remains a challenging task (especially in the absence of large in-domain labeled datasets) [8, 9]. Thus we use a word-level approach that offers both simplicity and flexibility. Further, this approach can act as a powerful baseline for more complex future methods.

We apply the UED framework to a corpus of English movie dialogues to quantitatively capture character emotion arcs. This not only helps us better understand the use of emotion words in character dialogues but also answer research questions pertaining to literary theory such as:

- To what degree do characters use emotion-associated words in the course of a narrative?

- How do characters differ in their reactivity to emotional events and their ability to recover from these events?

- Can we automatically identify key plot points through character dialogue? How do different characters react to key plot points.

- At what points in the story line are characters the least or most *emotionally discordant* with respect to other characters in some movie.

We present experiments to test hypotheses pertaining to these questions.

The contributions of this paper include: (1) introduce the concept of utterance emotion dynamics, (2) present metrics to capture utterance emotion dynamics, (3) use a simple emotion-lexicon based approach to calculate the metrics—an approach that can be easily applied to a large number of domains (without requiring labeled training data), and (4) test key hypotheses in literary theory.

Beyond literary studies, UED has applications in improving public health outcomes: e.g., tracking emotional arcs of patients in physiotherapy sessions, early detection of depression, and detecting emotional impacts of online harassment and hate speech. UED also has considerable potential in understanding the emotional underpinnings driving discourse and argumentation on social media: for example, how do people with different UED characteristics frame and react to arguments.

We begin with an overview of related work in NLP and Psychology. We then introduce the UED framework and metrics. Finally, we apply the UED metrics to analyze movie character dialogues, generate character emotion arcs, and test two hypotheses:

*hypothesis 1:* The amounts of negative emotions in utterances by characters tend to adhere to a systematic (non-random) trend over the course of a story (from beginning to end)

*hypothesis 2:* Character–character emotion discordances adhere to a systematic (non-random) trend over the course of a story.

Data, Code, and Ethics Statement associated with this project are be made available through the project webpage (https://github.com/whipson/edyn) The code includes an R package that

allows users to analyze their own data to determine UED metrics. The Ethics Statement is also included in the Appendix of this paper.

## Related work

UED addresses the importance of context and individual differences in emotions conveyed through their utterances. However, emotion dynamics is far from a mature science and its lack of operationalized concepts poses a substantial challenge. Our goal in this paper is to make these concepts concrete and to demonstrate how language can reveal aspects of an individual's emotion dynamics.

Work in Psychology on emotion dynamics often tends to assume a *dimensional* account of emotions, where an emotion state is described along two dimensions: *valence* (extremely unpleasant to extremely pleasant) and *arousal* (sluggish/sleepy to excited/activated) [10–12]. (Sometimes a third dimension, *dominance*, pertaining to weak vs. powerful, is also included [13]). In contrast, the *categorical* or *basic* account of emotions posits that there exist distinct emotion types (e.g., anger, joy, sadness) [14]. These models of emotion have had considerable influence since the 1960s, but lately have received significant criticism regarding their validity and universality [15–18].

Regardless of the underlying mechanisms of emotion, people often articulate their emotions through concepts such as anger, joy, sadness, and fear. Thus, we believe that studies involving utterance emotion dynamics can benefit from incorporating not just valence and arousal, but also frequently described emotions such as anger, joy, sadness, and fear. In this work we mainly explore UED in the valence–arousal space, but also consider the commonly studied categorical emotions for some metrics.

A common approach to analyzing literary texts (novels, plays, poetry, etc.) is to model the change in emotion words over whole texts [19–22]. Inferences can then be made as to whether stories adhere to prototypical "story shapes" or emotional arcs [6, 7, 23].

Some of the studies differentiating story characters based on their dialogue, include: distinguishing characters in terms of lexical style [24], exploring character networks [25], and character-character interactions [26]. More relevant to emotions, Jacobs [27] classified story characters along Big Five personality traits using emotion words and Rashkin et al. [28] used story characters' use of emotion words to infer build a system for detecting mental states (i.e., motivations, emotions, and goals). Klinger at al. [29] examined how emotion word usage changes over the course of a narrative for specific characters in a sample of German texts.

Our work extends these previous works in two main ways. First, we use an approach inspired by the psychological study of emotion dynamics to derive complex descriptors of how a character's emotions change over time, such as how quickly and how intensely their emotions deviate from their typical state. Second, we use a much larger sample of stories and characters than in previous work, using movie dialogue instead of literary text.

## Utterance emotion dynamics

We now introduce a series of UED metrics that are meant to be computed per person/character based on the emotion-associated words they utter. UED metrics can be extracted from a variety of data sources including social media posts (e.g., Tweets), historical speeches, and character movie dialogue (see next Section on Movie Dialogues) Input for computing these metrics is a sequence of words ordered in time or as per an individual's narrative text. Note that narrative text may not always be chronological. Narratives do not need to be continuous passages of text (i.e., they can be separated by breaks and interruptions), but they should consist of enough words so that reliable metrics can be derived from them. Examples of narratives

include the words a person utters over a week, all the words said by a literary character in a novel, and a character's dialogue in a movie.

1. **Emotion Word Density (EWD)**. *EWD* is the proportion of emotion words a person utters over a given span of time. One can determine emotion word density for individual emotion categories such as joy, sadness, etc. and for emotion dimensions such as valence ($v$) and arousal ($a$). In case of emotion dimensions, the valence (or arousal) scores act as weights for each word occurrence. Thus, emotion word density scores for valence and arousal are effectively the average valence and average arousal of the words, respectively.
Previous research has examined emotion word density in stories (averaging over all dialogue or text) [6, 7, 19, 21] and to track the flow of emotions in social media discourse [30, 31], but here we calculate density for each person.

2. **Home Base**. The *home base* is a subspace of high-probability emotional states for a person [32]. Analogously when analyzing utterances, one can consider emotion space (across one or more emotion dimensions). For example, at any given point in a character's narrative, their *location* in the valence–arousal space may be defined to be the point corresponding to the average valence and average arousal of the words uttered in some small window of recent utterances. We define the path traced by this location over time (as the narrative continues) as the *emotional arc* or *emotional trajectory* of the character. We define *home base* as the subspace where the character is most likely to be located. For example, in the one-dimensional case—e.g., either valence ($v$) or arousal ($a$)—the home base is the subspace pertaining to the most common average valence or arousal scores, respectively) [32]. We can mathematically define this band as the lower and upper bounds of a confidence interval:

$$\bar{v} \pm t_{(1-\alpha,N-1)} \sqrt{\frac{\sigma^2}{N}} \tag{1}$$

where $\bar{v}$ is the mean of $v$, $t$ is the t-distribution, $N$ is the number of components in $v$, $\alpha$ is the desired confidence (e.g., 68%—one standard deviation away from the mean), and $\sigma^2$ is the variance.
In the two dimensional valence–arousal space, the home base is bounded within an ellipse:

$$\frac{v_i - \bar{v}}{\psi \lambda_1} + \frac{a_i - \bar{a}}{\psi \lambda_2} = 1 \tag{2}$$

where $\bar{v}$ and $\bar{a}$ are the means for valence and arousal, $\psi$ is the critical $\chi^2$ value for the desired confidence range (e.g., 68%—one standard deviation) and $\lambda_1$ and $\lambda_2$ are the eigenvalues of the covariance matrix. The values in the denominator of the two terms correspond to the major and minor axes (i.e., the two diameters). The coordinates $v$ and $a$ that make the equality true are the boundaries of the ellipse and any set of coordinates that make the expression $< 1$ are within the ellipse. We can then compare home base ellipses among different people in terms of their *location* (in the valence-arousal space) and their size in terms of the ellipses' major and minor axes (see Fig 1).

3. **Emotional Variability**. *Emotional variability* is the extent to which a person's emotional state changes over time. We follow the approach offered by [33] and measure variability as the standard deviation (SD):

$$\text{SD}(v) = \frac{\sum_{i=1}^{N} (v_i - \bar{v})^2}{N} \tag{3}$$

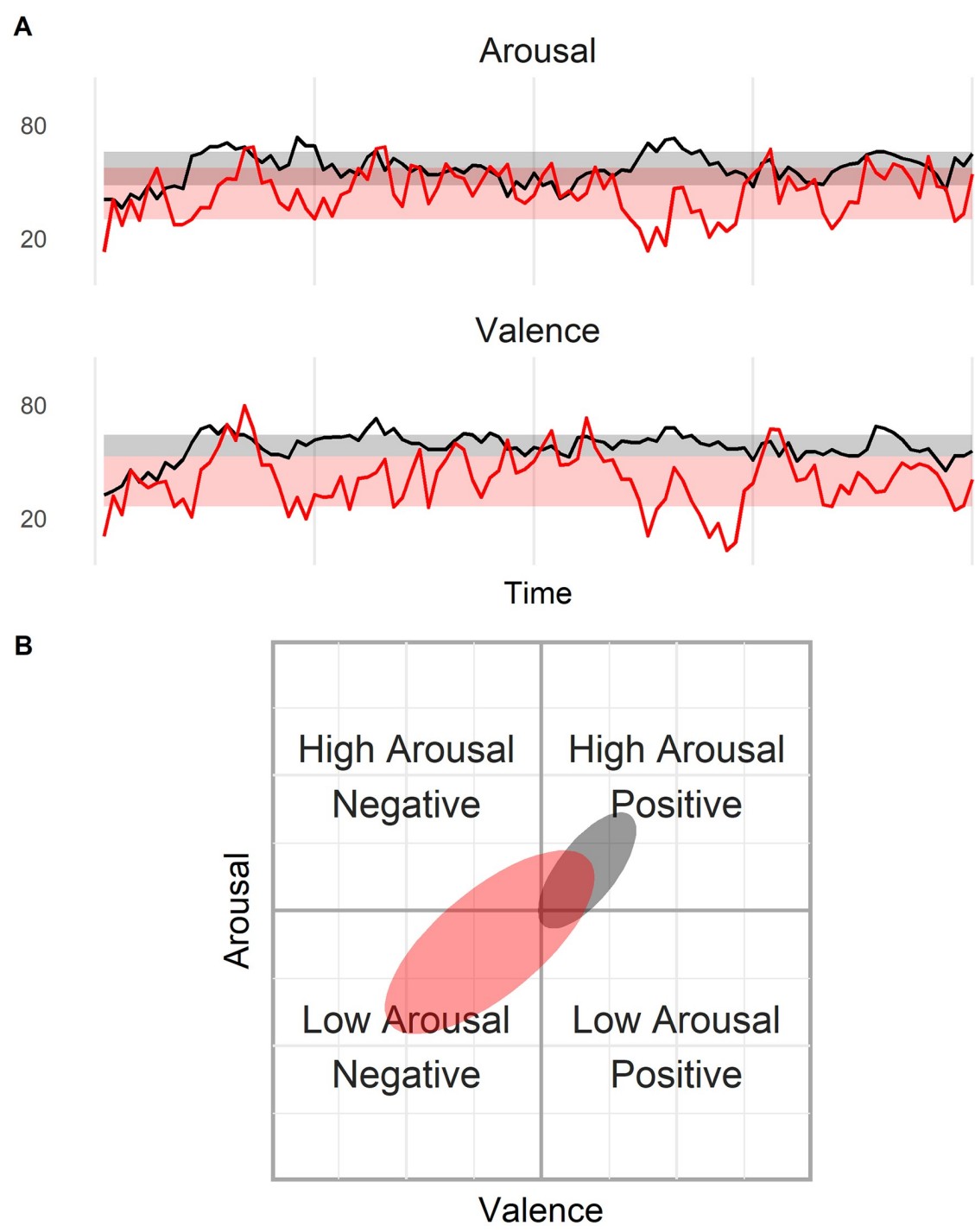

**Fig 1. Panel A (one dimensional)**: Two hypothetical (simulated) emotion trajectories —separately for valence (v) and arousal (a). Shaded bands are the home bases. **Panel B (two dimensional)**: Home bases of two hypothetical trajectories in v–a space.

In the two dimensional case, variability is defined as the average of the SDs of *v* and *a*.

4. **Displacement Count and Length**. At times, one may use more or less emotion words than usual, which results in a movement from inside the home base to outside the home base. We refer to this phenomenon as *displacement*. We define a person's *displacement count* as

the number of times in their narrative that they deviate from their home base. We define *displacement length* as the number of words uttered from the point the person left their home base to the point they returned (a proxy for how long the person was away from home base). We average these to obtain the *average displacement count* and *average displacement length*, respectively.

5. **Peak Distance**. For each displacement, we define its peak as the point furthest from the home base, and *peak distance* as how far away the point is from the home base (in terms of Euclidean distance from the perimeter of the home base). The choice of Euclidean distance instead of another distance measure such as cosine distance was motivated by our conceptualization of displacement as *movement* around a two-dimensional state space. We average the peak distances for each person's displacements to obtain the *average peak distance*.

6. **Rise and Recovery Rates**. Rise rate is the rate at which a person reaches peak emotional intensity and recovery rate is the rate at which a person returns to their home base. Rise rate can be seen as an indicator of emotional reactivity [34, 35]. Emotional reactivity is the speed and intensity with which a person responds to emotional situations [35]. Recovery rate can be seen as an indicator of emotion regulation. [36, 37]. Emotion regulation is a person's ability to return to their typical emotional state [36]. These rates are computed by dividing the peak distance by the number of words during the rise or recovery period, respectively. For example, if the peak is not far from the home base, but many words are uttered before reaching the home base, then the recovery rate value is a small number, indicating slow rate of recovery. Fig 2 shows displacements for two hypothetical individuals. The person denoted by the black line has a much slower recovery rate compared to the red

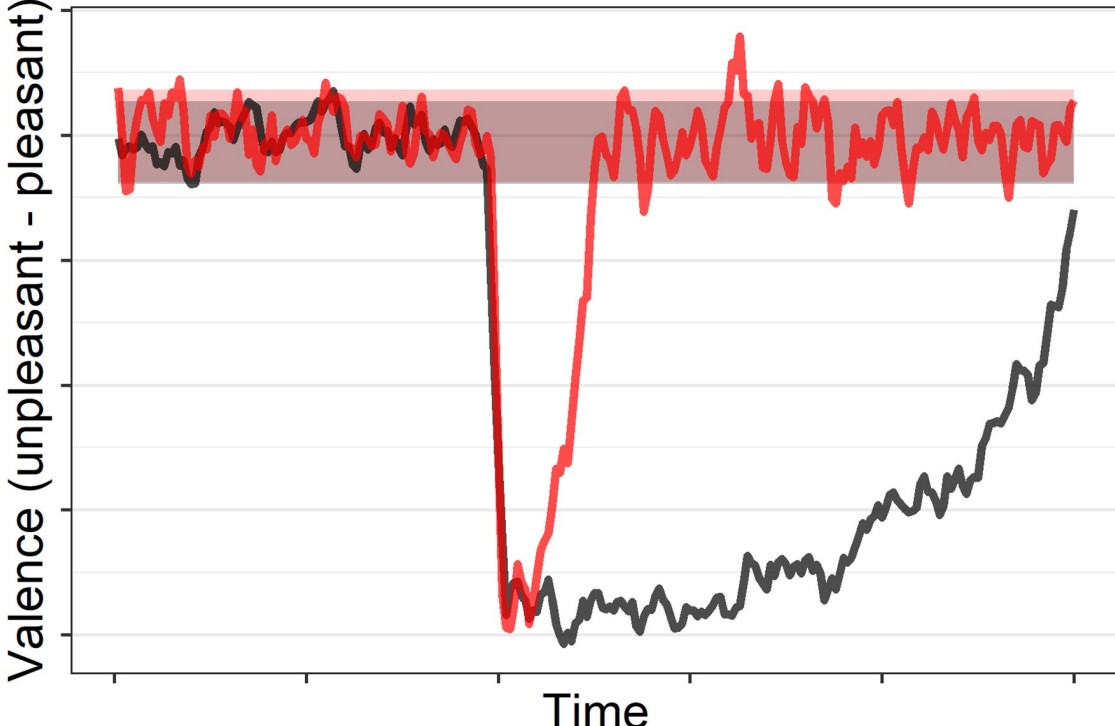

**Fig 2. Change in valence over time for two hypothetical (simulated) people (black and red).** The black line depicts a much slower recovery rate than the red line.

line. We average the rates for a person's displacements to obtain average rise and recovery rates.

Together the metrics (1 through 6) robustly capture temporal emotional characteristics of a person's utterances. Despite the simplicity of these metrics, no previous work in NLP has attempted to so closely align the theory of emotion dynamics with applications in NLP and digital humanities.

## Emotion dynamics in movie dialogues

We apply the UED framework and metrics discussed above to a movie dialogue corpus to determine temporal emotion characteristics in the utterances of characters.

The Sections below describe the resources used, the UED metrics associated with the characters in the movie corpus, and how we tested two hypotheses pertaining to character emotion arcs: (1) There will be systematic trends over the course of characters' narratives in the amount of negative emotion utterances; (2) Character-character emotion discordances will adhere to a systematic trend over the course of a story.

### Data

We briefly describe below the movie dialogues corpus and the emotion lexicons used.

**IMSDb corpus.** We accessed scripts from the Internet Movie Script Database (IMSDb) during the month of February 2020 (https://www.imsdb.com/) This website curates a database of movie scripts and allows free access to them for non-commercial purposes. We began with 1,210 movie scripts; then removed 83 for formatting issues (e.g., unconventional dialogue markers), and another 4 because they were non-English scripts. Finally, there remained 1,123 movie scripts with 54,518 characters.

The dialogues within the scripts are identified and grouped into turns. We define a *turn* as a sequence of uninterrupted utterances by a character. In other words, a turn begins when a character's dialogue stops and ends when either a different character's dialogue starts or the movie ends. For our experiments, we considered characters that had at least 50 turns in a movie. There were 2,687 such characters (roughly 5%). We will refer to them as the *main characters*.

We processed the text using the WordNet Lemmatizer [38] and an off-the-shelf tokenizer [39]. This left us with 5,673,201 word tokens, an average of 5,102 tokens per movie, and 1,376 tokens per character.

**Emotion lexicons.** We used two existing manually curated word–emotion association lexicons to determine the UED metrics: the NRC Emotion Lexicon [40, 41] (freely available at http://saifmohammad.com/WebPages/NRC-Emotion-Lexicon.htm) and the NRC Valence-Arousal-Dominance (NRC VAD) Lexicon [42] (freely available at http://saifmohammad.com/WebPages/nrc-vad.html). The NRC Emotion lexicon contains about fourteen thousand commonly used English words and their associations with eight basic emotions (*anger, anticipation, disgust, fear, joy, sadness, surprise,* and *trust*) and two sentiments (*negative* and *positive*). The NRC VAD lexicon contains about twenty thousand commonly used English words that have been scored on *valence* (0 = extremely unpleasant, 1 = extremely pleasant), *arousal* (0 = extremely sleepy/sluggish, 1 = extremely activated/excited), and *dominance* (0 = extremely powerful, 1 = extremely weak). As an example, the word *nice* has a valence of.93 and an arousal of.44. (We do not make use of the dominance scores here, but those can be explored in future work. Future work can also explore the intensity of emotions over narrative time, using

**Table 1. Average emotion word density (Av. EWD) and standard deviation (SD) of main characters in IMSDb ($N$ = 2,687).**

| Emotion | Av. EWD | SD |
|---|---|---|
| Negative | 16.5 | 3.8 |
| Positive | 20.3 | 4.5 |
| Anger | 7.6 | 2.6 |
| Anticipation | 12.0 | 2.8 |
| Disgust | 5.6 | 2.3 |
| Fear | 9.9 | 3.0 |
| Joy | 9.8 | 3.5 |
| Sadness | 8.3 | 2.5 |
| Surprise | 6.8 | 2.0 |
| Trust | 13.5 | 3.4 |

lexicons such as the NRC Affect Intensity Lexicon [43] (available at: http://saifmohammad.com/WebPages/AffectIntensity.htm).

Note that the lexicons themselves provide only likely emotion associations and do not take into account the context of neighboring words in the target text. Nonetheless, since most words have a highly dominant primary sense [44], and the metrics capture emotion associations from a large number of words, this simple approach is effective.

## Utterance emotion dynamics of movie characters

We calculated the UED metrics discussed above for the 2,687 main characters (with more than 50 turns each) in the IMSDb Corpus using existing emotion lexicons. Tables 1 and 2 summarise average metrics for characters in the IMSDb. These numbers serve as benchmarks allowing one to compare a character's emotion word usage with what is common across movies.

1. **Emotion Word Density.** Table 1 shows the average density of emotion words uttered per character. The rows correspond to the individual emotion categories. Observe that characters use an average of 20.3% positive words compared to 16.5% negative words. We note as well that characters use trust (13.5%), anticipation (12.0%), and joy (9.8%) words more than disgust (5.6%), surprise (6.8%), and anger (7.6%) words.

2. **Home Base.** We calculate metrics 2 through 6 on valence and arousal scores because these are continuous. We arrange each character's words in temporal order and apply a 10-word

**Table 2. Average UED metrics (2–6) and standard deviation (SD) in the v–a space for main characters in IMSDb ($N$ = 2,687).**

| Metric | Av. UED | SD |
|---|---|---|
| Home Base-Major Width | 0.13 | 0.02 |
| Home Base-Minor Width | 0.09 | 0.01 |
| Emotion Variability | 0.15 | 0.02 |
| Displacement Length | 9.13 | 1.90 |
| Displacement Count | 34.46 | 18.01 |
| Peak Distance | 0.17 | 0.03 |
| Rise Rate | 0.05 | 0.01 |
| Recovery Rate | 0.05 | 0.01 |

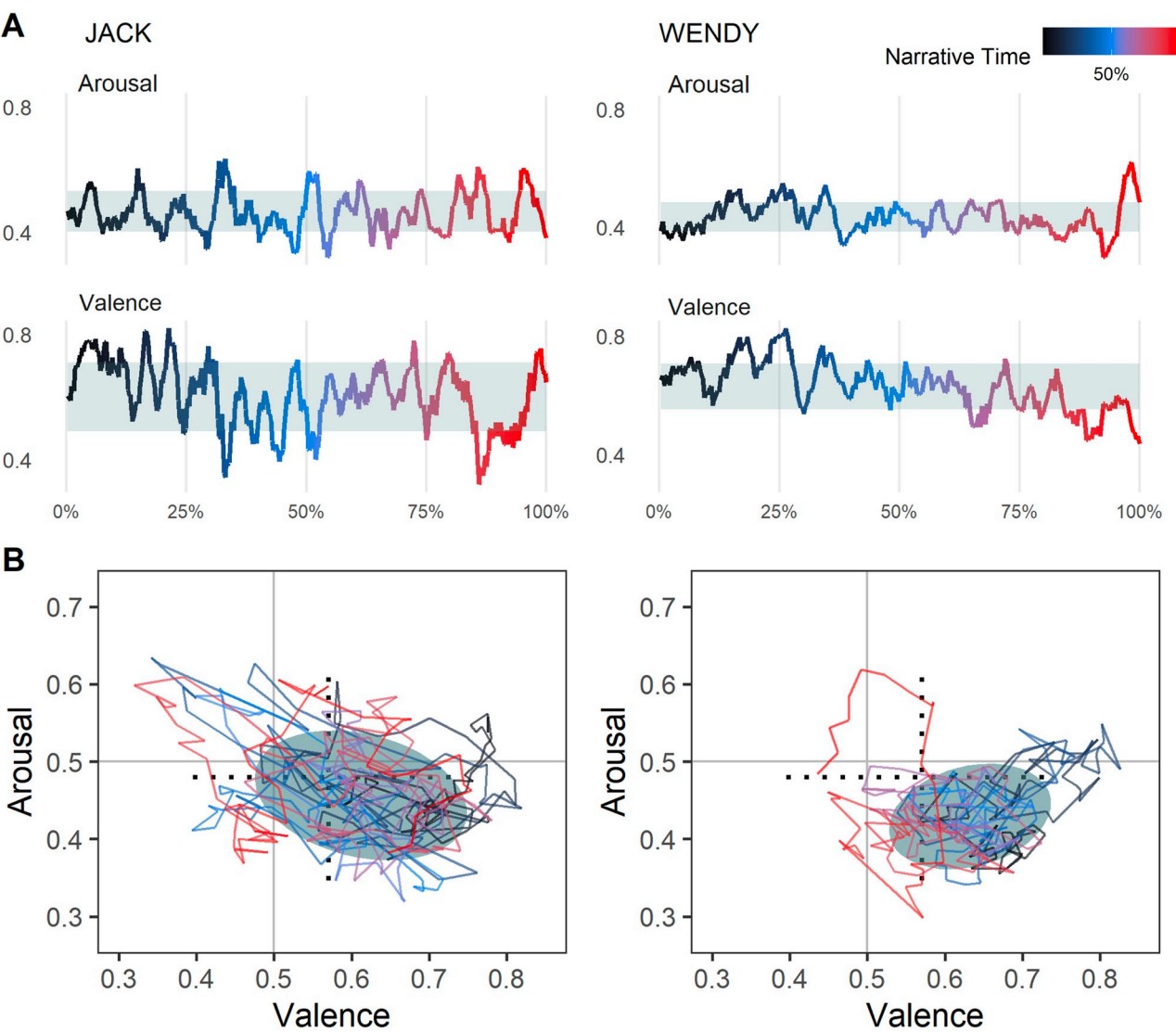

**Fig 3. One dimensional and two dimensional state spaces for Jack (*n* = 389 words) and Wendy (*n* = 279 words), two main characters from The Shining (1980).** Color of line corresponds to narrative time, with dark blue meaning earlier in the movie and red meaning later. The black dotted lines show the major and minor axes of an ellipse within which all main characters are 95% of the time (the ellipse itself is not shown to avoid clutter).

rolling average to both the valence and arousal scores. We include only those words that are present in the VAD lexicon.

We computed the home bases for all the main characters (*N* = 2,687). Shaded regions in Panels A and B of Fig 3 show the home bases for two main characters in *The Shining (1980)*. (Synopsis: *Jack and his family move into an isolated hotel with a violent past. Jack begins to lose his sanity, which affects his family members [Wendy and Danny]*). Observe that Jack's home base is wider (semi-major axis = 0.159 vs. 0.123) than Wendy's but roughly the same in terms of height (semi-minor axis = 0.115 vs. 0.111).

3. **Variability.** From Fig 3 it is evident that Jack is more emotionally variable than Wendy. Using Eq 3, we can quantify the emotional variability (EV): EV(Jack) = 0.166, EV(Wendy) Wendy = 0.135. The supplementary material shows the characters with the highest and

lowest variability. The least emotionally variable character is Data from Star Trek. In contrast, among the most variable is Ginger, from the movie Casino, who is described as "cunning and manipulative. . . ne of the greatest female movie villains". (https://villains.fandom.com/wiki/Ginger_McKenna)

4. **Displacement Length.** We found that, on average, a character experiences 34.46 displacements (departures from home base) with a standard deviation of 18.01. We also observe that the average displacement length is 9.13 VAD words (#words uttered between leaving the home base and returning to it) with a standard deviation of 1.90 words. To put this in perspective, the average turn contains 3.27 VAD words, suggesting that the typical displacement lasts roughly three turns.

   We find that Jack has 31 displacements compared to Wendy's 19. However, Jack's displacements are shorter on average (9.30 words) than Wendy's (11.14 words). Fig 4 shows an example of one displacement from Jack's narrative.

5. **Peak Distance.** Characters tend to peak at an average distance of 0.166 (in the v–a space) away from their home base. Jack's average peak distance (0.206) is nearly twice as large as Wendy's (0.106).

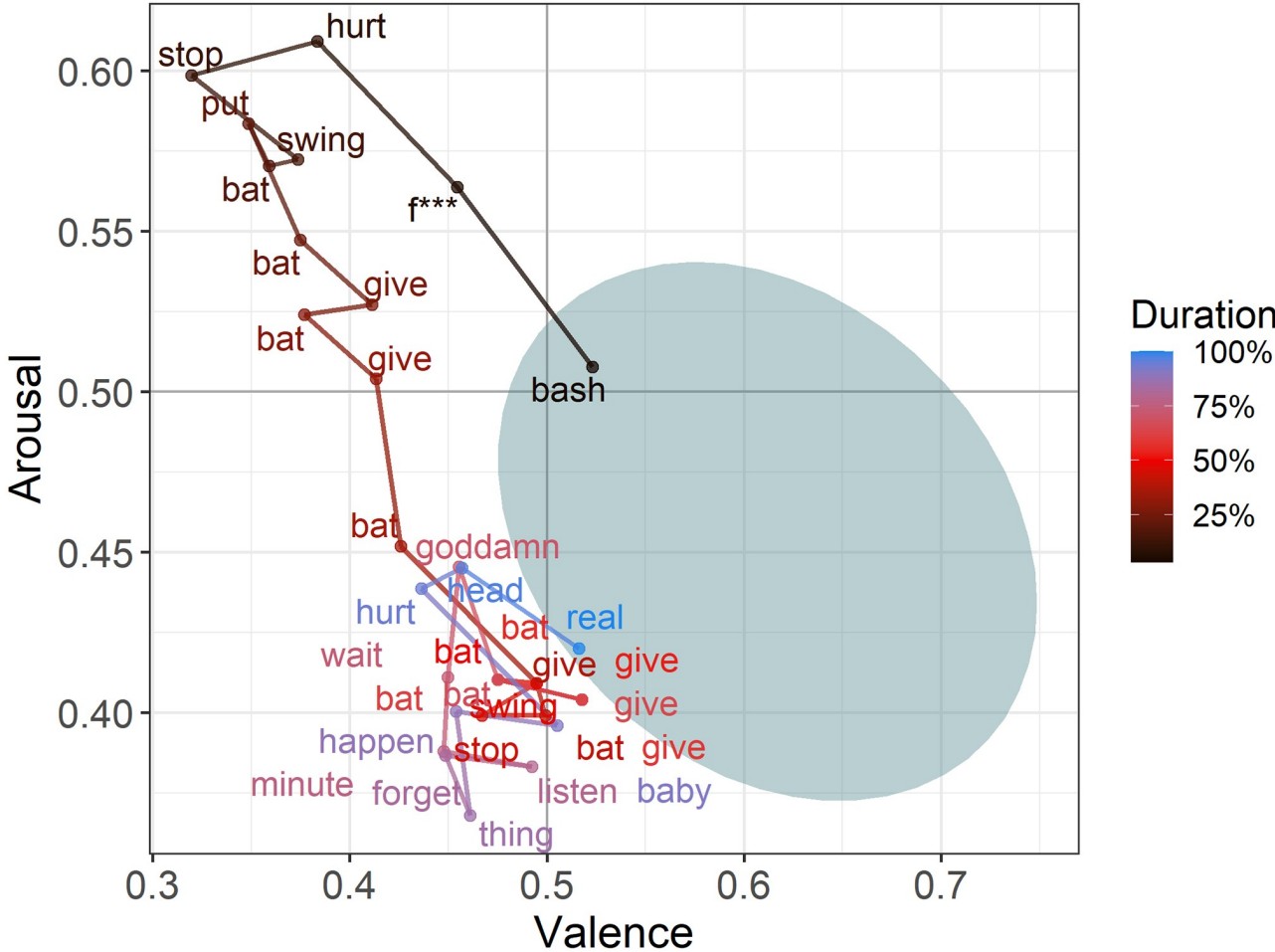

**Fig 4. Example displacement with underlying words (from Jack's dialogues in The Shining).** Note: Here, the location of a word does not correspond to its v{a score, but rather to the character's v{a rolling average when the word was uttered.

**Table 3. Characters with the highest/lowest emotional variability (var.) in the v–a space.** Note that the bottom rank number is less than the total number of characters in the data because some characters had insufficient number of displacements to obtain reliable averages.

| Rank | Character | Movie Title | Var. |
|---|---|---|---|
| 1 | Jessica | Little Athens | 0.228 |
| 2 | TJ | Hesher | 0.220 |
| 3 | Ginger | Casino | 0.215 |
| 4 | Dennis | Hostage | 0.208 |
| 5 | Wes | Three Kings | 0.204 |
| 2610 | Lynn | L.A. Confidential | 0.107 |
| 2611 | Diane | Horse Whisperer | 0.104 |
| 2612 | Dolores | Sweet Hereafter | 0.103 |
| 2613 | Riker | Star Trek | 0.103 |
| 2614 | Data | Star Trek | 0.100 |

6. **Rise and Recovery Rate.** The average rise rate across all characters is 0.051 as is the average recovery rate, meaning that the average character travels a distance of 0.051 per word during the rise period and recovery period. Jack has a higher rise rate (0.059) compared to Wendy (0.023) and a higher recovery rate (0.064) compared to Wendy (0.027).

Tables 3 and 4 show the top five and bottom five characters in terms of emotion variability and recovery rate (two UED metrics), respectively. We note that one of the characters with the highest variability, Ginger from *Casino*, is a highly noted movie villain, while the character with the lowest variability is Data from *Star Trek*—an android who supposedly experiences minimal emotion.

**Trend in individual character arcs.** Literary studies explore several research questions about narrative arcs. The UED metrics described earlier can be used to characterize emotional arcs of individual characters, which in turn can be used to inform our broader understanding of how characters arcs and stories are composed. In this and the next section, we explore specific hypotheses pertaining to trends in individual character emotion arcs and trends in character–character interactions, respectively.

**Table 4. Characters with highest/lowest recovery rate (rec.) in the v–a space.** Note that the bottom rank number is less than the total number of characters in the data because some characters had insufficient number of displacements to obtain reliable averages.

| Rank | Character | Movie Title | Rec. |
|---|---|---|---|
| 1 | Jacob | Nightmare on Elm Street | 0.107 |
| 2 | Chad | Burn After Reading | 0.106 |
| 3 | Andrew | The Breakfast Club | 0.105 |
| 4 | Rennie | Friday the 13th | 0.101 |
| 5 | Jimmy | Magnolia | 0.101 |
| 2610 | Jonson | Anonymous | 0.019 |
| 2611 | Agnis | Shipping News, The | 0.018 |
| 2612 | Paul | Manhattan Murder Mystery | 0.017 |
| 2613 | Jack | Burlesque | 0.017 |
| 2614 | Chigurh | No Country for Old Men | 0.015 |

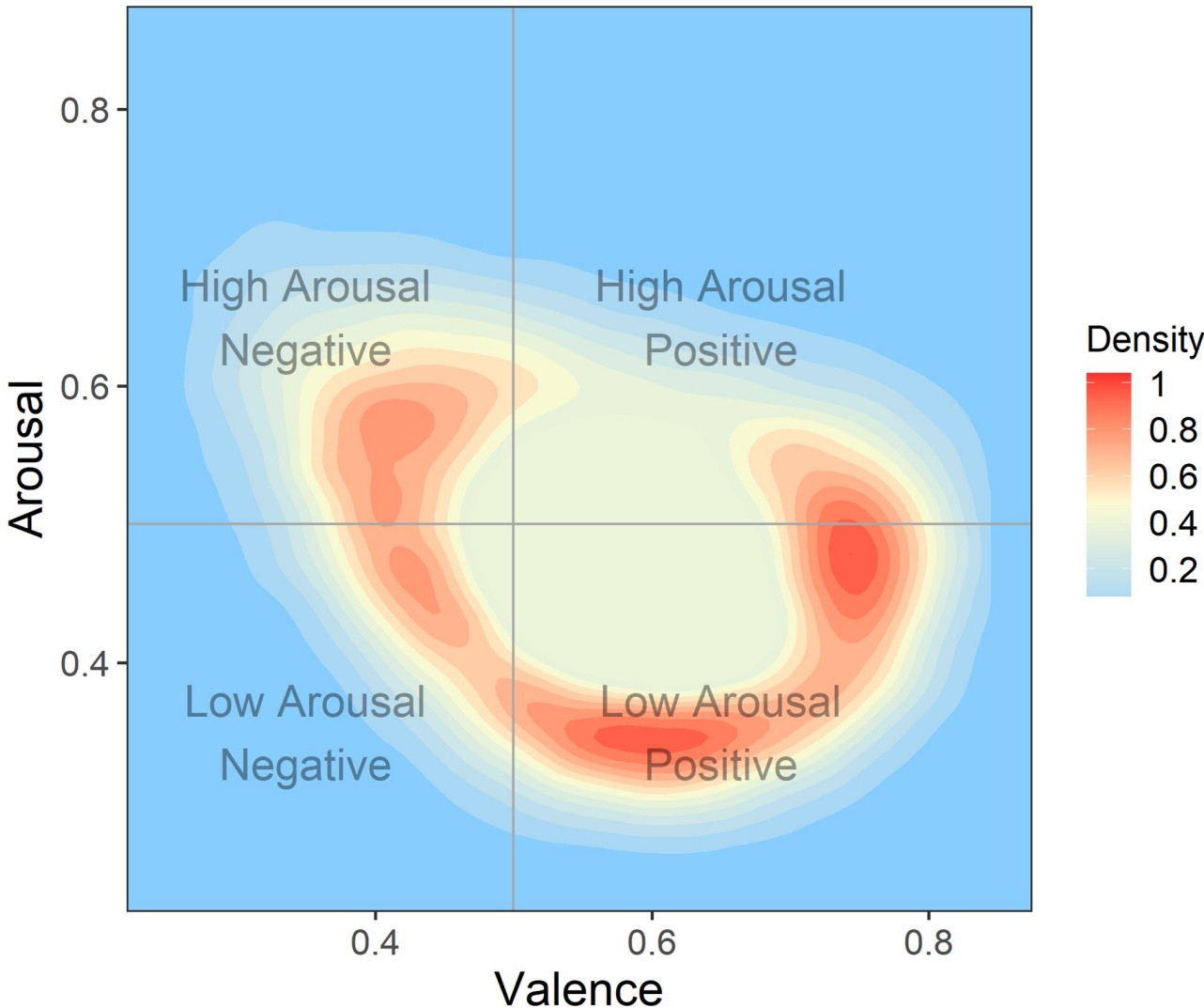

**Fig 5. Density map showing where peak displacements tend to occur.** Red corresponds to more peaks. Density is normalized to go from 0–1.

A compelling question in the literary analysis of individual characters is whether there are commonalities in the emotional arcs of story characters? This can be explored in terms of patterns in the emotion space and in terms of how the emotions change over time.

*Emotion space.* We first explore the question: Where in the v–a space do characters tend to experience displacements? We do so by creating a topological map of peak displacement location and frequency with which a location was the point of peak displacement (see Fig 5). We find that peaks tend to occur near the average home base (i.e., most displacements peak at a short distance), and in two regions in particular: low arousal positive (feeling content) and moderate arousal positive (feeling happy). There is a shorter peak in the high arousal negative (feeling agitated).

*Time.* Next we explore: What is the shape of the average emotion arc? Is the average arc essentially a flat line? Does it monotonically increase/decrease in certain emotions with time? Is the average arc wavy with many peaks and troughs? etc. One hypothesis is that characters become increasingly negative toward the end of the narrative as the plot reaches its climax. We implemented a 30-word rolling average for each character's positive and negative word density

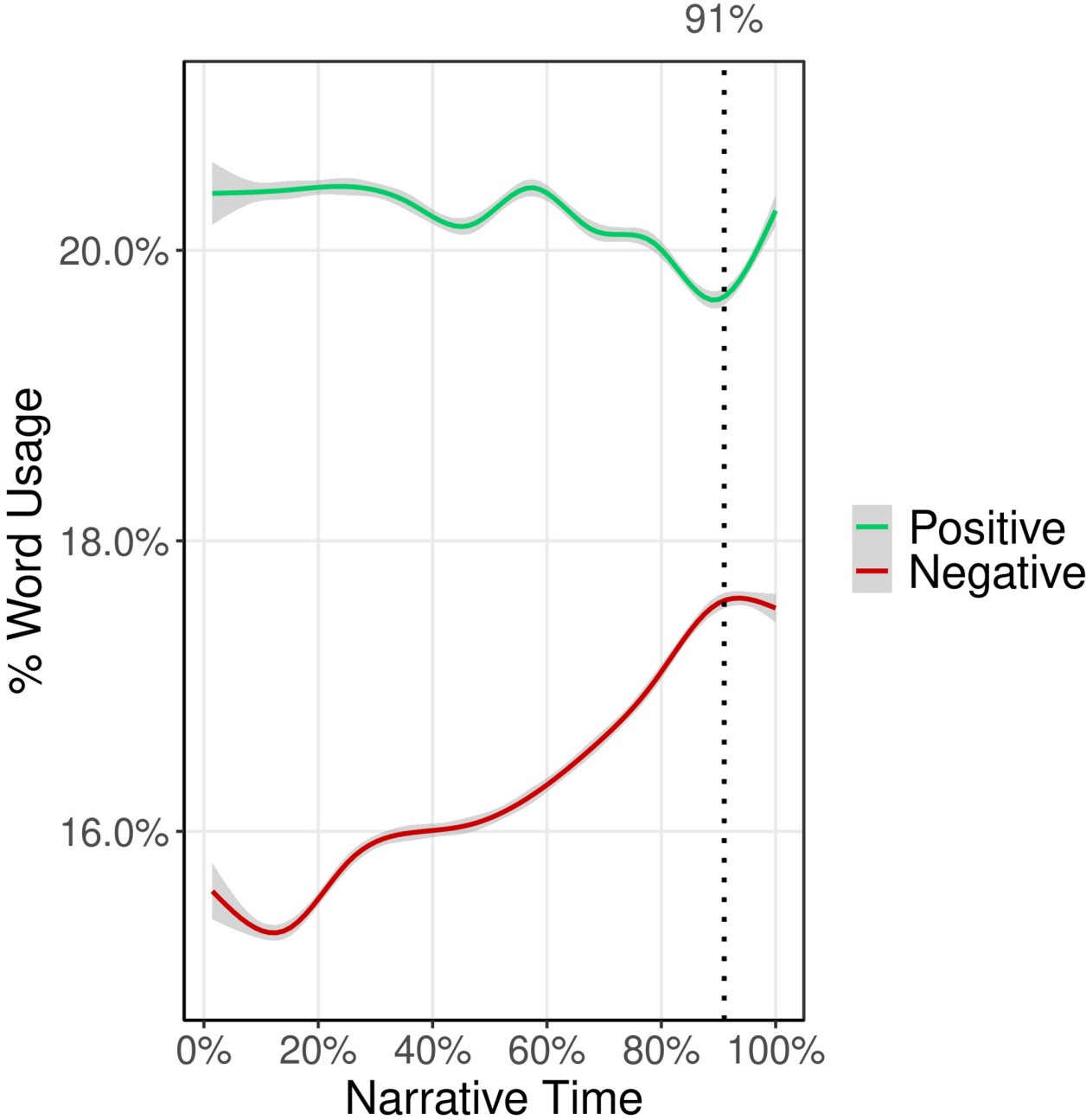

**Fig 6. Average trends in proportion of positive and negative word usage by characters across narrative time.** Vertical dotted line shows location of peak negative density and lowest positive density. Grey band is the 95% confidence interval around the estimated mean. *n* = 965,147 words.

and normalized the narrative time of each character's dialogue. We then regressed positive and negative density on narrative time. Specifically, we used a Generalized Additive Model—a penalized spline regression—to allow for curved relationships between narrative time and density [45].

*Results.* We found that narrative time was significantly ($p < 0.001$) associated with positive and negative density, suggesting that positive and negative density do vary systematically over the narrative arc. Fig 6 shows the shape of these trends across all characters for the positive and negative word density.

We found that, on average, negative words uttered by characters increase in frequency by about 2% over the duration of a movie, peaking at 91% of the duration of a movie. We also observe a less pronounced, but clearly observable, decline in the use of positive words over the course of a movie, also reaching the lowest density at about the 91% mark. After the 91% mark, there is a reversal of the trends, probably because the conflicts in a story begin to resolve.

Since characters may have varying spans over which they appear in a movie, and some characters only appear later or early in the movie, we repeated the above experiments on a subset of characters that are present in the beginning 10% and the final 10% of the stories ($N$ = 2,151). These experiments also showed similar patterns as we see in Fig 6.

We also constructed emotion arcs for the eight basic emotions and found that "negative" emotions like anger and fear followed a similar pattern to negative sentiment shown in Fig 6 (same was true for "positive" emotions like joy and trust).

**Trends in character–character interactions and discordance.** Another set of research questions is around how the emotion arcs for different characters in the same movie change with narrative time (possibly due to the interactions between the characters and a result of the events in the story). Some past work has explored character emotional conflict [46], but only in the context of a small number of stories.

At any given point, pairs of characters may use emotion words similarly (e.g., both use lots of low valence words), or dissimilarly (e.g., one uses high-arousal words whereas the other uses low-arousal words). We will refer to these as *in-sync* and *discordant* pairs. Of potential interest to literary theorists are questions such as: to what extent do character–character discordances vary throughout the movie plot? Are there some trends common across movie plots regarding when character–character discordances tend to be the lowest and when they tend to be highest? etc. Similar to the emotion word density arcs, we hypothesize that the average discordance tends to peak near the end of the story.

We applied a similar approach to determine character–character emotion distance as was used to calculate displacement length (distances from a character's home base), only now we consider distances from other character's emotional states. We limited the following analysis to only include characters who appeared during the first 10% and last 10% of the movie so that they would have approximately the same narrative length. Discordances were calculated for all possible pairs of characters (who met the inclusion criteria) within each movie.

*Results*. A multilevel model showed that narrative time was significantly ($p < 0.001$) and positively associated with narrative time, suggesting that discordance increases over narrative time.

Fig 7 shows the average character–character discordance over the course of a movie. We see that discordance is relatively low toward the beginning, steadily increases, and is highest during the last 10% of the narrative. This suggests that toward the end of the film characters tend to be further apart in their emotion word usage. The difference between peak and lowest discordance is a distance of 0.02 in the v–a space, but it is worth noting that characters tend to occupy a smaller subspace (see crosshairs on Fig 3). 0.02 is about 6% the crosshair length along the valence dimension and about 9% along the arousal dimension.

Average character–character discordance peaks at $\sim$90% of narrative length. Interestingly, this is roughly in the same region where average negative word density peaks (91%). To explore this further, we computed the correlation between the two. (Discordance and negative word density are not required to be correlated: theoretically, discordance could increase because the characters move in emotionally opposite directions). We found that character–character discordance correlates positively with negative density ($r = 0.691$) and correlates negatively with positive density ($r = -0.623$). We interpret this as: characters on average become

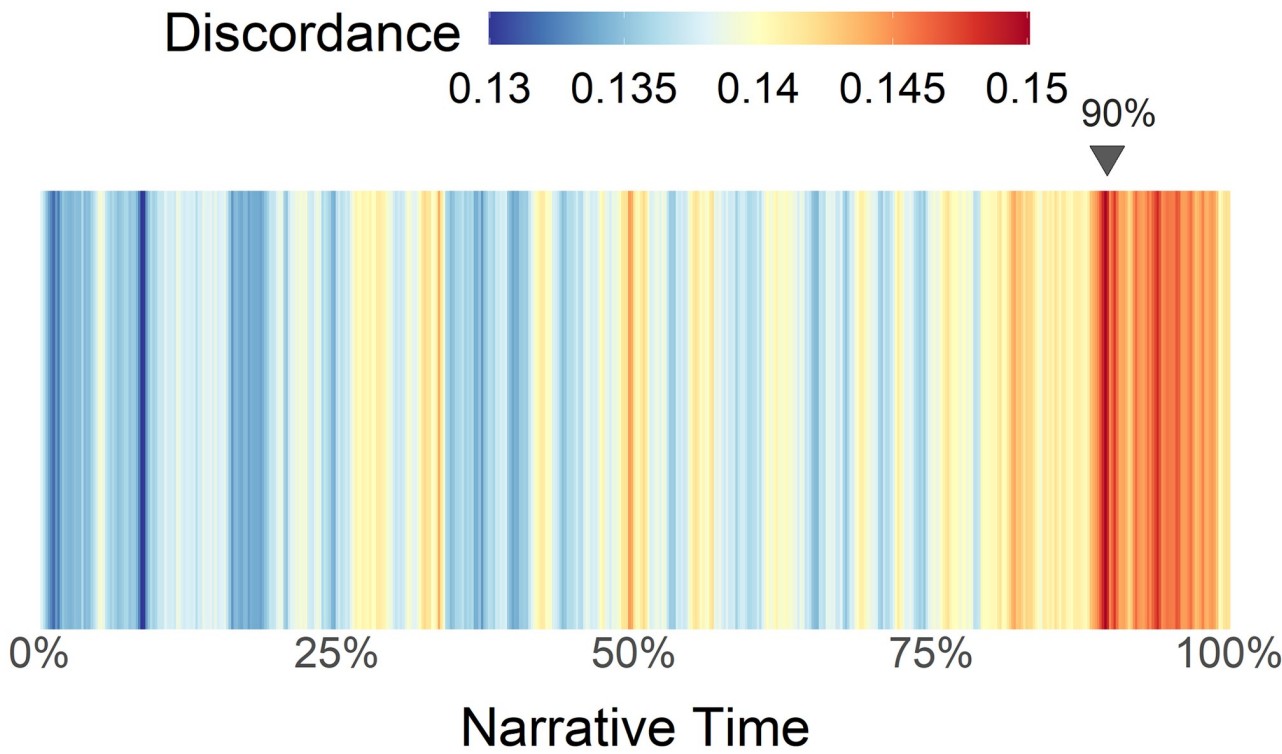

**Fig 7. Average character–character discordance over narrative time ($n$ = 1,246,990 words).** Red indicates more discordance, blue indicates less discordance. Discordance is lowest during first quarter of a movie and peaks at 90%. Score is measured in the same scale as the v-a space (i.e., 0.15 implies a Euclidean distance of 0.15 in the state space).

more negative over time but some characters also become increasingly discordant with others over time, perhaps increasing in negative emotion at a faster rate than other characters in the same movie.

## Conclusions

Building on the literature in Psychology, we introduced a framework to track the dynamics of an individual's emotion word usage over time. Specifically we outlined several utterance emotion dynamics metrics. We used a corpus of movie dialogues to compute individual and average UED metrics for thousands of characters. These numbers serve as benchmarks for the analysis of characters. We analyzed the emotional arcs to show that: on average, negative words uttered by characters increase over the course of a movie, peaking at 91% of the narrative length; the maximum character–character discordance is observed roughly in the same region; and that negative word density and character–character discordance are correlated.

The results raise new questions as to why such trends exist in the emotional arcs of characters and what purpose they serve in story telling. This is an active area of future research for us. We are also interested in developing the UED metrics further, applying them to other corpora (social media discussions, transcripts of psychotherapy sessions, etc.), and explore hypotheses pertaining to the interrelations of one's emotion dynamics and their behavior. We see applications of utterance emotion dynamics not just in NLP but also in social sciences, psychology, public policy, and public health.

We freely release the data and code associated with the project. Users can thus easily determine UED metrics on data of interest. We also provide an Ethics & Data Statement that

outlines the intended uses of the methods and data introduced here, their limitations, and cautions about their future use—especially on drawing mental health inferences about individuals without their consent and without clinical data.

## Appendix

### Ethical considerations in the use of UED metrics

In this paper 'Emotion Dynamics in Movie Dialogues', we introduced utterance emotion dynamics (UED) metrics and applied these metrics to dialogues of movie characters. However, UED metrics can be used on other data as well. Below we list key ethical considerations in the use of UED metrics. Please see the main paper for details about Utterance Emotion Dynamics and the movie dialogues corpus. Please see the papers associated with the emotion lexicons for information about how the lexicons were created and their intended uses. Applying UED metrics to any new data, should only be done after first investigating the suitability of such an application, and requires care to ensure that it produces the desired results and minimizes unintentional harm. See Mohammad [47] for a general set of ethical considerations in the use of emotion lexicons. Notable issues especially relevant to this work are listed below:

1. **Inferences About Mental Health:** Do not draw inferences about mental health or personality traits from UED metrics for an individual without meaningful consent and without corresponding clinical data. Future work is planned (in collaboration with clinicians and health experts) for the use of UED metrics to help with public health.

2. **Inferences for Aggregate-level Analysis vs. Individuals:** Even though UED metrics are focused on individuals, their benefit and reliability are greater when UED metrics from many individuals are analyzed together to identify broad patterns in how emotions change over time. Do not draw inferences about individuals (e.g., hiring suitability or job performance prediction) from UED metrics. We do not expect these metrics to be reliable indicators for that. We do not think such use is ethical. More generally we recommend use of UED metrics only after consent of the individuals whose text is being used. If applying to public data, we stress again, the importance of aggregate-level analysis as opposed to drawing inferences about individuals.

3. **Coverage:** UED metrics make use of word–emotion association lexicons. However, even the largest lexicons do not include all terms in a language. The high-coverage lexicons, such as the NRC Emotion Lexicon and the NRC VAD Lexicon have most common English words. The lexicons include entries for mostly the canonical forms (lemmas), but also include some morphological variants. However, when using the lexicons in specialized domains, one may find that a number of common terms in the domain are not listed in the lexicons.

4. **Dominant Sense Priors:** Words when used in different senses and contexts may be associated with different emotions. The entries in the emotion lexicons are mostly indicative of the emotions associated with the predominant senses of the words. This is usually not too problematic because most words have a highly dominant main sense (which occurs much more frequently than the other senses). When using the lexicons in specialized domains, one may find that a term might have a different dominant sense in that domain than in general usage. Such entries should be removed before using the lexicon in the target domain.

5. **Associations/Connotations (not Denotations)**: A word that denotes an emotion is also associated with that emotion, but a word that is associated with an emotion does not

necessarily denote that emotion. For example, *party* is associated with joy, but it does not mean (denote) joy. The lexicons capture emotion *associations*. Some have referred to such associations as *connotations or implicit emotions*. The associations do not indicate an inherent unchangeable attribute. Emotion associations can change with time, but these lexicon entries are largely fixed, and pertain to the time they are created or the time associated with the corpus from which they are created.

6. **Inappropriate Biases:** Since the emotion lexicons have been created by people (directly through crowdsourcing or indirectly through the texts written by people) they capture various human biases. Some of these biases may be rather inappropriate. For example, entries with low valence scores for certain demographic groups or social categories. For some instances, it can be tricky to determine whether the biases are appropriate or inappropriate. Capturing the inappropriate biases in the lexicon can be useful to show and address some of the historical inequities that have plagued humankind. Nonetheless, when these lexicons are used in specific tasks, care must be taken to ensure that inappropriate biases are not amplified or perpetuated. If required, remove entries from the lexicons where necessary.

7. **Relative (not Absolute):** The absolute values of the UED scores themselves are not as useful as how the scores compare to the rest of the relevant population. For example, knowing that a person has negative word density score of 0.34 is not very useful on its own, but knowing that this score is more than two standard deviations higher than the average for the relevant population, can be useful.

**Pro-tip**:

1. Manually examine the emotion associations of the most frequent terms in your data. Remove entries from the lexicon that are not suitable (due to mismatch of sense, inappropriate human bias, etc.).

## Author Contributions

**Conceptualization:** Will E. Hipson, Saif M. Mohammad.

**Data curation:** Will E. Hipson, Saif M. Mohammad.

**Formal analysis:** Will E. Hipson.

**Investigation:** Will E. Hipson.

**Methodology:** Will E. Hipson.

**Project administration:** Saif M. Mohammad.

**Resources:** Saif M. Mohammad.

**Software:** Will E. Hipson.

**Supervision:** Saif M. Mohammad.

**Validation:** Will E. Hipson.

**Visualization:** Will E. Hipson.

**Writing – original draft:** Will E. Hipson.

**Writing – review & editing:** Will E. Hipson, Saif M. Mohammad.

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
