## [Decision Letter · Decision Letter 0]

28 Apr 2021

PONE-D-21-06829

Emotion Dynamics in Movie Dialogues

PLOS ONE

Dear Dr. Hipson,

Thank you for submitting your manuscript to PLOS ONE. After careful consideration, we feel that it has merit but does not fully meet PLOS ONE’s publication criteria as it currently stands. Therefore, we invite you to submit a revised version of the manuscript that addresses the points raised during the review process.

We look forward to receiving your revised manuscript.

Kind regards,

Haoran Xie

Academic Editor

PLOS ONE

Journal Requirements:

3. Please modify the title to ensure that it is meeting PLOS’ guidelines (https://journals.plos.org/plosone/s/submission-guidelines#loc-title). In particular, the title should be "specific, descriptive, concise, and comprehensible to readers outside the field" and in this case it is not informative and specific about your study's scope and methodology.

*PLOS ONE has specific criteria for papers that describe new methods, databases, or software for applications. Specifically, these reports must meet the criteria of utility, validation, and availability, which are described in detail at http://journals.plos.org/plosone/s/submission-guidelines#loc-methods-software-databases-and-tools. To that effect, please address in the manuscript whether the UED metrics proposed here represent a proven advantage over alternative tools, should they exist.

Reviewers' comments:

Reviewer's Responses to Questions

**Comments to the Author**

1. Is the manuscript technically sound, and do the data support the conclusions?

Reviewer #1: Yes

Reviewer #2: Yes

2. Has the statistical analysis been performed appropriately and rigorously? 

Reviewer #1: Yes

Reviewer #2: I Don't Know

3. Have the authors made all data underlying the findings in their manuscript fully available?

Reviewer #1: Yes

Reviewer #2: Yes

4. Is the manuscript presented in an intelligible fashion and written in standard English?

Reviewer #1: Yes

Reviewer #2: Yes

5. Review Comments to the Author

Reviewer #1: The work proposed a new metric, "Utterance Emotion Dynamics", to study emotion-associated words in movie scripts. Following are few comments:

1. I think it would be better to move data description part in front of method section. So the readers would have a clear picture about how the values are generated.

2. Applied methods should be written in one section, for example, the hypothesis testing.

2. Table captions should be at the top of the tables.

3. Lines 24 and 25 are not needed.

4. in Table 2, please define what is "displacement count".

5. Figure 4, please explain what is "the character's location in the v-a space".

6. Line 261, typo: "be explore"?

7. Figures 1 and 2, the presented two characters are chosen randomly? Please explained.

8. Figure 6, it would be better to add S.D. or 1-, 3-quartile in addition to average, to display the variations.

9. About the character-character discordances, is this done by all possible pairs or selected? Please clarify.

10. Lines 187-188, "a turn as a sequence of uninterrupted utterances by a character". What is "uninterrupted", a character continues to talk, without stops? Do scenes change count as interruption?

11. Result section should be made more clear.

Reviewer #2: Great subject, which might attract specific interest in various sciences such as neuroscience, psychology, forensic and psychotherapy and performing arts in addition to the relevant industries. Considering recent pandemic which has promoted various meeting via visual technologies, this piece of research would be warmly greeted.

It would be interesting to see the further similar studies by the authors and other interested scientists.

6. PLOS authors have the option to publish the peer review history of their article (what does this mean?). If published, this will include your full peer review and any attached files.

Reviewer #1: No

Reviewer #2: **Yes: **Dr Lily Abedipour MD

---

## [Author Response · Author response to Decision Letter 0]

16 Jun 2021

Official Response Letter

Dear editor and reviewers. Thank you for your thoughtful comments. We have made several changes to address them. Details are below (Reviewer comments are italicized):

Reviewer 1 comments:

1. I think it would be better to move data description part in front of method section. So the readers would have a clear picture about how the values are generated.

Because the metrics can be applied to a variety of data sources, we opted to introduce the metrics first followed by the description of the movie scripts data as the application of the metrics. However, we have included a statement at the beginning of the method (Section: Utterance Emotion Dynamics) explaining that UED metrics can be extracted from a variety of data sources. We also include a pointer to the section specifically describing the Movie Dialogues data.

2. Applied methods should be written in one section, for example, the hypothesis testing.

Because of the novelty of the UED metrics, we structured the paper in a way to build complexity and show how the UED metrics can be applied to answer a variety of interesting questions. The first set of results focus on how we can use UED to describe characters’ emotion dynamics. The second set of results focuses squarely on testing hypotheses about wider trends across stories and character interactions. We think this structure helps the reader understand how to use the UED metrics to answer increasingly complex questions about emotion dynamics.

2.  Table captions should be at the top of the tables.

Fixed.

3. Lines 24 and 25 are not needed.

These lines are now a footnote.

4. in Table 2, please define what is "displacement count".

We have added a sentence in the UED metrics section more clearly defining displacement count and length.

5. Figure 4, please explain what is "the character's location in the v-a space".

Done. We added a sentence describing that it is the character’s v-a rolling average when the word was uttered.

6. Line 261, typo: "be explore"?

Fixed.

7. Figures 1 and 2, the presented two characters are chosen randomly? Please explained.

Done. We clarify that the two characters chosen were two main characters.

8. Figure 6, it would be better to add S.D. or 1-, 3-quartile in addition to average, to display the variations.

Done. We added 95% confidence bands.

9. About the character-character discordances, is this done by all possible pairs or selected? Please clarify.

We added a sentence clarifying that discordances were calculated for all possible pairs of characters (who met the inclusion criteria) within each movie.

10. Lines 187-188, "a turn as a sequence of uninterrupted utterances by a character". What is "uninterrupted", a character continues to talk, without stops? Do scenes change count as interruption?

We added the following sentence to make this clearer: In other words, a turn begins when a character’s dialogue begins, and ends when either a different character’s dialogue starts or the movie ends. Scene changes do not count as an interruption if the same character speaks at the end of one scene and at the start of the next scene. (This is okay here because we continue to track the emotional state of the character through their utterances before and after scene change.) 

11. Result section should be made more clear.

We made the following changes to improve the results section:

 1. We reduced the amount of reiteration of related work and interpretation that is covered in earlier or later sections. The idea is to keep the results section more clearly focused on the actual findings, while leaving the bulk of contextual information for other sections.

 2. We streamlined the reporting of statistical results. We omit overt references to the null hypotheses as these are often clear from the context of the statistical test.

 3. Some tangential statistical tests have been made into footnotes. For example, in the hypothesis test of peak discordance, we save the test conducted on the subset of characters present at the beginning and end 10% of the narrative for a later footnote.

Reviewer #2 comments: 

Great subject, which might attract specific interest in various sciences such as neuroscience, psychology, forensic and psychotherapy and performing arts in addition to the relevant industries. Considering recent pandemic which has promoted various meeting via visual technologies, this piece of research would be warmly greeted. It would be interesting to see the further similar studies by the authors and other interested scientists.

Thank you for the support and encouragement.  We have included some connections of this work, especially with psychology and public health, but we welcome suggestions on specific ways to lnk this work to the various fields you have mentioned.

---

## [Decision Letter · Decision Letter 1]

2 Aug 2021

Emotion Dynamics in Movie Dialogues

PONE-D-21-06829R1

Dear Dr. Hipson,

We’re pleased to inform you that your manuscript has been judged scientifically suitable for publication and will be formally accepted for publication once it meets all outstanding technical requirements.

Kind regards,

Haoran Xie

Academic Editor

PLOS ONE

Additional Editor Comments (optional):

The paper is ready for publication.

Reviewers' comments:

Reviewer's Responses to Questions

**Comments to the Author**

1. If the authors have adequately addressed your comments raised in a previous round of review and you feel that this manuscript is now acceptable for publication, you may indicate that here to bypass the “Comments to the Author” section, enter your conflict of interest statement in the “Confidential to Editor” section, and submit your "Accept" recommendation.

Reviewer #1: All comments have been addressed

2. Is the manuscript technically sound, and do the data support the conclusions?

Reviewer #1: (No Response)

3. Has the statistical analysis been performed appropriately and rigorously? 

Reviewer #1: (No Response)

4. Have the authors made all data underlying the findings in their manuscript fully available?

Reviewer #1: (No Response)

5. Is the manuscript presented in an intelligible fashion and written in standard English?

Reviewer #1: (No Response)

6. Review Comments to the Author

Reviewer #1: (No Response)

7. PLOS authors have the option to publish the peer review history of their article (what does this mean?). If published, this will include your full peer review and any attached files.

Reviewer #1: No

---

## [Editor Report · Acceptance letter]

2 Sep 2021

PONE-D-21-06829R1 

Emotion Dynamics in Movie Dialogues 

Dear Dr. Hipson:

I'm pleased to inform you that your manuscript has been deemed suitable for publication in PLOS ONE. Congratulations! Your manuscript is now with our production department. 

Kind regards, 

on behalf of

Professor Haoran Xie 

Academic Editor

PLOS ONE